# The Influence of Keratinized Mucosa on the Nonsurgical Therapeutic Treatment of Peri-Implantitis

**DOI:** 10.3390/jcm11144118

**Published:** 2022-07-15

**Authors:** Carla Fons-Badal, Rubén Agustín-Panadero, Carlos Labaig-Rueda, Maria Fernanda Solá-Ruiz, Marina García-Selva, Antonio Fons Font

**Affiliations:** Department of Oral Medicine, Faculty of Medicine and Dentistry, University of Valencia, C/Gascó Oliag, 1, 46010 Valencia, Spain; carla.fons@uv.es (C.F.-B.); labaig@uv.es (C.L.-R.); m.fernanda.sola@uv.es (M.F.S.-R.); marina.garcia@uv.es (M.G.-S.); antonio.fons@uv.es (A.F.F.)

**Keywords:** dental implants, peri-implantitis, keratinized mucosa, peri-implant tissues, non-surgical treatment

## Abstract

Objective: The main objective of this study aimed to evaluate the response to treatment in implants with peri-implantitis regarding the presence or absence of keratinized mucosa. Materials and Methods: One hundred implants with peri-implantitis were treated non-surgically at the Prosthodontics and Occlusion Teaching Unit of the University of Valencia. Records were registered at day zero (recession, bleeding on probing, suppuration, probing depth, bone loss and plaque index), at 1, 3 and 6 months. Results: In both groups, a slight increase in recession and disappearance of suppuration took place, and no bone loss was observed during the following 6 months. However, after an initial reduction, probing depth, bleeding on probing and plaque index increased again in the group without keratinized mucosa (KM). Conclusions: Implants with KM presented better results after non-surgical treatment as well as in their medium-term evolution, although it would be advisable to increase the number of samples in order to achieve greater scientific evidence and standardization in the treatment protocol. Clinical relevance: The role of keratinized mucosa in the development of peri-implantitis has been mentioned in many publications, but less has been emphasized regarding its influence on the success of the treatment of this pathology. The presence of keratinized mucosa has been found to play a key role in the evolution of the non-surgical treatment of peri-implantitis. The amount of keratinized mucosa should be considered in the treatment of peri-implantitis, as well as when planning connective tissue grafts accompanying the decontamination of implants in the absence of keratinized mucosa.

## 1. Introduction

Implant treatment has recently become a frequent treatment option in order to replace missing dental pieces. However, over the years, the presence of both mechanical and biological complications has been repeatedly observed. Therefore, innovative materials and technologies have become intense research topics in the dentistry field in order to improve treatment outcomes and reduce morbidity as well as biological and surgical times [1]. Peri-implantitis has become a relatively frequent biological complication, with a range of 1–47% [2], and an important one as well, since it can lead to the loss of the implant. 

Treatment of peri-implantitis focuses on decontamination of the implant surface and conditioning of the surrounding tissues. A former non-surgical phase should always be performed, as it allows for evaluating the tissue response and the patient’s hygiene [3]. Many authors endorse that this first non-surgical phase is generally effective in resolving peri-implantitis [4,5,6,7], although the effectiveness of non-surgical treatment has been a controversial subject, with a diversity of opinions throughout the literature [2,8].

Various factors can influence the occurrence of peri-implantitis such as a history of periodontitis, tobacco consumption, diabetes, inadequate plaque control, absence of regular maintenance, absence of keratinized mucosa (KM), genetic factors, systemic conditions, iatrogenic factors, etc. [9,10,11]. Such factors may also influence the long-term evolution of peri-implantitis treated implants. 

Hence, the main purpose of this study was to assess the response to treatment in implants with peri-implantitis in relation to the presence or absence of keratinized mucosa. 

The working hypothesis proposed stated that implants that are not surrounded by keratinized mucosa present worse health maintenance in the medium term.

## 2. Materials and Methods

A prospective study was carried out in order to evaluate the response to treatment of 100 implants (from the brand Biomet 3i (Palm Beach, FL, USA)) with peri-implantitis, placed in the Prosthodontics and Occlusion Teaching Unit of the University of Valencia. The implants included in the study were selected during the control visits carried out throughout the year 2021. It was imperative that they had been loaded with their definitive prosthesis for at least one year, presented clinical signs of peri-implantitis following the criteria indicated in the 2017 Workshop [10] and had an adequate general periodontal health status. 

Initial records related to the patient were collected (age, gender, tobacco consumption habits and history of periodontitis), having placed all implants in patients already treated and periodontally stabilized [12]; as well as other features related to the implants: Millimeters of keratinized mucosa measured in the mid-zone of the buccal site in implants using a periodontal probe CP 15 by Hu-friedy^®^ (Chicago, IL, USA); Recession: millimeters of gingival margin retraction with respect to the prosthetic crown (where the gingival margin was located at the time of prosthesis placement) measured in the mid-buccal site; Silness and Löe plaque index; Bleeding and suppuration: its presence or absence was measured. Probing depth using the CP15 periodontal probe (at six points: mesiobuccal, buccal, distobuccal, mesiopalatal, palatal and distopalatal); Gingival inflammation (Löe and Silness Gingival Index); and percentage of bone loss: measured using the Rhinoceros software (Robert Mcneel & Associates. Seattle, WA, USA) on radiographs taken with Rinn system positioners (Denstsply, Charlotte, IL, USA). The years since prosthesis placement were also recorded. All data were collected by a single operator.

Once the initial records were obtained, a non-surgical decontamination treatment was performed under local anesthesia (articaine 4% and adrenaline 1:100,000) which consisted of the removal of plaque and calculus from the implant surface using ultrasound (SP Newtron, Satelec Acteon, La Marnasse, Olliergues, France) and H3 tip (Satelec Acteon, Olliergues, France), implant-specific titanium curettes by Hu-friedy^®^ (Chicago, IL, USA), curettage of the internal epithelium of the pocket with Hu-friedy^®^ 4R/4L curette (Chicago, IL, USA), subgingival irrigation with an antiseptic (chlorhexidine 0.2%), administration of antibiotic (metronidazole 500 mg every 8 h/7 days) and antiseptic (0.2% chlorhexidine gel 3 times a day for 15 days). Supragingival scaling treatment was also performed in the rest of the oral cavity and oral hygiene instructions were provided. There were no patients allergic to the drugs administered, therefore, all of them received the same treatment. 

After completing the treatment, a re-evaluation was carried out at 1 month, 3 months and 6 months to assess the outcome of the treatment, in which the same data as at baseline were collected to analyze both the initial results and the medium-term evolution (Table 1). In the evaluation at one month, it was also determined whether the patient should go on to maintenance or whether a previous surgical phase was required. 

Statistical analysis was performed with the data obtained.

### 2.1. Ethics and Consent to Participate

The study was approved by the Human Research Ethics Committee of the University of Valencia with procedure number 183748 and all participants signed a written informed consent.

### 2.2. Statistical Analysis 

The main objective of the statistical analysis was to describe the clinical evolution of implants with a diagnosis of periimplantitis that receive treatment and its association with KM. Simple binary logistic regression models were estimated using generalized estimating equations (GEE) to explain the probability of implant recession, bleeding and suppuration on probing or inflammation of the implant depending on the presence, or absence of KM over time. 

Likewise, for clinical variables of ordinal or continuous nature (probing depth, bone loss, plaque), generalized linear models were also applied under the GEE approach. This type of approach is justified by the multiplicity of implants per patient, thus controlling the dependence of the observations.

The models were replicated by adjusting the time since implant placement and the amount of KM (no presence, ≤2, >2 mm). In order to study the association between the presence of KM and the pre-treatment clinical situation, the above methodology was also employed.

The significance level used in the analyses was 5% (α = 0.05). Due to the multilevel design of the data (several implants per patient), assuming a high intra-subject correlation (ρ = 0.75), a power of 89% was obtained. The statistical software used was SPSS 15.0. 

## 3. Results

A total of 100 implants with peri-implantitis were analyzed in 25 patients (17 females and 8 males), with a mean age of 64.6 ± 6.3. The mean time of implant loading was 12.27 ± 5.6. From the total number of subjects analyzed, 21 presented a previous history of periodontitis versus 4 that did not, suggesting a strong relationship between a previous history of periodontitis and the development of peri-implantitis.

The sample was divided into two groups of implants based on the presence or absence of KM. In the records obtained at baseline (Pre-study T0), 51.0% of the implants presented with KM, with a mean KM bandwidth of 1.43 ± 1.68 mm, and 49.0% presented an absence of KM (Figure 1). The remaining implant characteristics at baseline are shown in Table 2. 

At one-month follow-up (T1), three implants had to be explanted due to poor prognosis and evolution. From this moment on, the entire analysis was performed on 97 implants. 

When analyzing the evolution of the initial parameters after treatment, a slight (not statistically significant) increase in recession was observed at the one-month follow-up (T1), but it remained stable (*p* = 0.151) in the subsequent follow-ups (T2 and T3), both for implants with and without KM (*p* = 0.773) (Figure 2). 

The presence of bleeding underwent significant changes throughout follow-up. These changes appeared to be associated with the presence of KM (*p* < 0.001). Bleeding was substantially reduced at one month (T1) in both groups, but eventually reappeared over time, being more pronounced as the amount of KM present decreased (*p* < 0.001), with rates exceeding 72% in the absence of KM (Figure 3). 

Treatment practically eliminated suppuration (*p* < 0.001), being equally effective with and without KM (*p* = 0.218) and this situation was sustained throughout the follow-ups. 

Probing depth, which was initially deeper in the absence of KM (5.16 mm ∅ KM (absence of keratinized tissue) vs 4.43 mm in the KM group), was reduced after treatment in both groups (T1: 3.93 mm ∅ KM vs 3.44 mm KM group), but thereafter, those implants with KM maintained a stable probing depth and those without KM increased once again (T2: 4.09 mm ∅ KM vs 3.46 mm KM group and T3: 4.38 mm ∅ KM vs 3.55 mm KM group). At a descriptive level, there seemed to be a different evolutionary pattern in both groups (Figure 4) and these differences were statistically significant (*p* < 0.001).

Regarding bone loss, significant differences were detected at T0: the smaller the quantity of KM surrounding the implant, the greater quantity of bone loss was experienced: 46.78% without KM vs 29.97% in presence of KM (OR = 0.96; *p* = 0.001). Thereafter, all clinical assessments were stable, with no bone alterations observed throughout the follow-up period.

The bacterial plaque index was initially higher in implants without KM (OR = 0.44; *p* = 0.027). After treatment, there was a similar reduction in both groups (T1) that seemed to be maintained at 3 months. However, the descriptive impression was that in implants without KM it tended to rebound at 6 months, while it continued to decline in implants with KM. The model concludes that plaque will progress again more readily in the total absence of KM (*p* = 0.007) (Figure 5).

## 4. Discussion

Over the years, non-surgical treatment of peri-implantitis has been considered a limited option when it comes to solving peri-implant infection [3]. Thanks to technological advances and treatment evolution, new research suggests an improvement in the results previously presented [7,13,14]. Roos-Jansaker obtained a reduction of bleeding on probing from 97% to 38%, remaining stable at 3 months, and Nart pointed out a reduction of probing depth to less than 5 mm in 95.45% and none presented progression of bone loss. The present results are in line with these studies, with a decrease in bleeding on probing from 100% to 12.4% after initial treatment, a statistically significant reduction in probing depth from 4.78 mm to 3.68 mm and no progression of bone loss. 

Regarding the non-surgical treatment sequence, no standard protocol exists, but it varies according to the studies; the most standardized one being the mechanical decontamination of the implants and the administration of antiseptics [15]. The intake of systemic antibiotics also seems to improve outcomes [16,17]. In the present study, decontamination was performed by ultrasound and curettage, irrigating with 0.2% chlorhexidine and Metronidazole 500 mg every 8 h for 7 days [7]. 

In the reevaluation, approximately one month after the initial decontamination treatment, it is decided if the implant continues with supportive therapy or if a surgical phase is required because there is no control over the infection signs. The choice of a regenerative technique will depend on the characteristics of the patient and the conditions surrounding the implant [18]. Non-surgical therapy should always be performed before any surgical intervention since it grants time to evaluate the healing response and the patient’s ability to perform effective oral hygiene measures [3]. If infection signs reappear during the periodontal implant support therapy, retreatment should be considered. 

Based on previous knowledge of periodontal infections, it is apparent that modifications in clinical conditions are necessary to prevent a recurrence of the pathology. One of the clinical conditions that should be conveniently altered, besides hygiene, is the amount of KM [19]. KM, as well as being related to the aesthetics of the peri-implant tissues, is also related to their health.

In the research presented (already at baseline) the absence of KM was associated with a poorer clinical and radiographic status of the implant. Therefore, the effectiveness of the treatment must be evaluated by comparing the evolution of the two groups (with and without KM) and not the final values at 6 months. KM seems to influence the medium-term stability of many of the clinical signs of peri-implantitis. The presence of pre-treatment recession was significantly higher without the presence of KM but remained stable in both groups throughout the visits. Treatment significantly eliminated bleeding at 1 month but increased in the absence of KM in the medium term. Suppuration was similar in both groups. It was eliminated after treatment and remained stable. Probing depth was greater in the absence of KM; the reduction was of a similar extent in both groups, but subsequently rebounded in the absence of KM while remaining stable in its presence. In the pre-treatment phase, the plaque index was higher in the absence of KM, subsequently reduced by one point on average, although a recurrence was observed in the group without KM. Finally, inflammatory signs increased from the third month in implants without KM. At the beginning of the treatment, all clinical signs improved; however, over the months, in many of the implants without KM, they returned to levels similar to the initial ones. This seems to indicate that the presence of KM intervenes in the medium and long-term stability of peri-implant health. After non-surgical treatment, the results in the group with KM remained more stable, probably because the KM band allows greater tissue stability due to surface keratinization and better access to hygiene. Therefore, KM not only has an aesthetic role around the implants but also seems to be a protective factor, improving the results of non-surgical treatment of peri-implantitis and making them more stable in the medium term. 

However, there are current studies still pointing out that the role of KM in peri-implantitis has low scientific evidence and needs further research [20,21].

## 5. Conclusions

Implants with the presence of KM obtain better results after non-surgical treatment of peri-implantitis as well as during their evolution and stability in the medium term, emphasizing the stability in the reduction of bleeding, probing depth and plaque index.

It is convenient to standardize the protocol for non-surgical peri-implant treatment. 

Although the present study is supported by other works in the literature, it would be advisable to increase the number of samples in order to achieve greater scientific evidence.

## Figures and Tables

**Figure 1 jcm-11-04118-f001:**
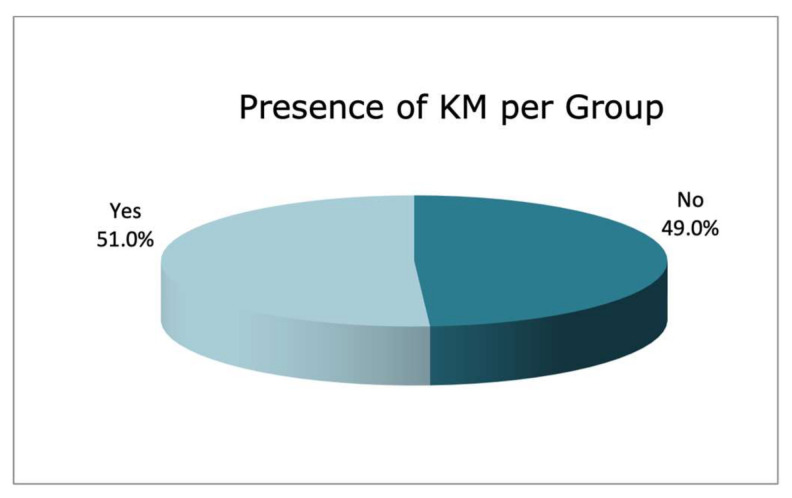
Presence of KM.

**Figure 2 jcm-11-04118-f002:**
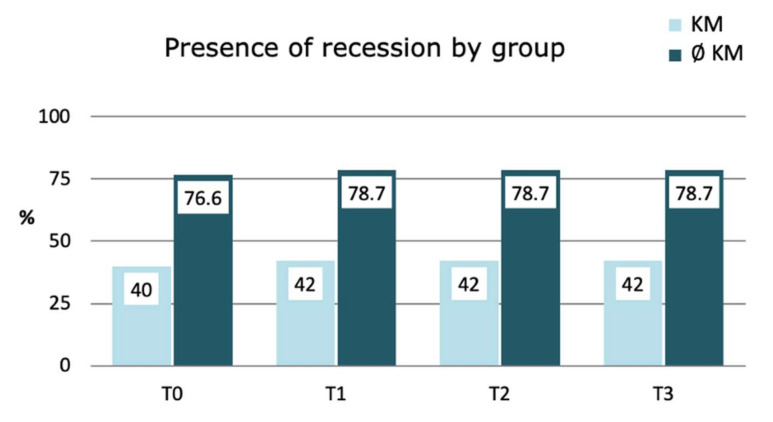
Variations in the prevalence of recession according to KM Group: Results for simple logistic regression with GEE model.

**Figure 3 jcm-11-04118-f003:**
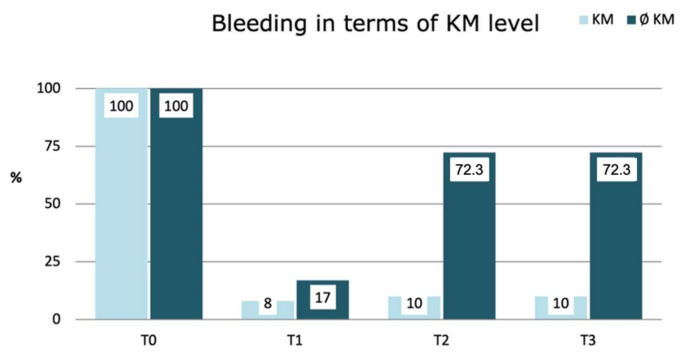
Changes in bleeding prevalence according to KM Level: Simple logistic regression results with GEE model.

**Figure 4 jcm-11-04118-f004:**
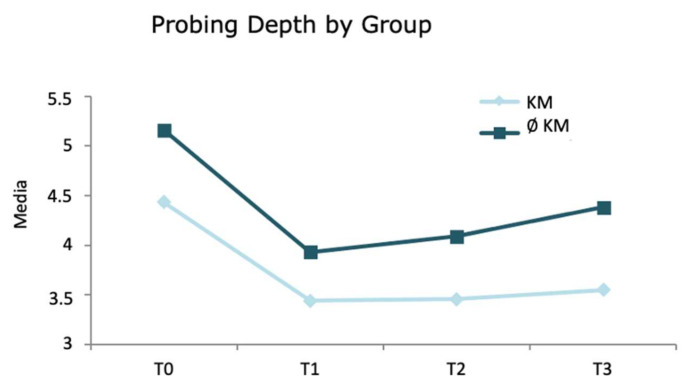
Changes in probing depth according to KM Group: Simple linear regression results with GEE model.

**Figure 5 jcm-11-04118-f005:**
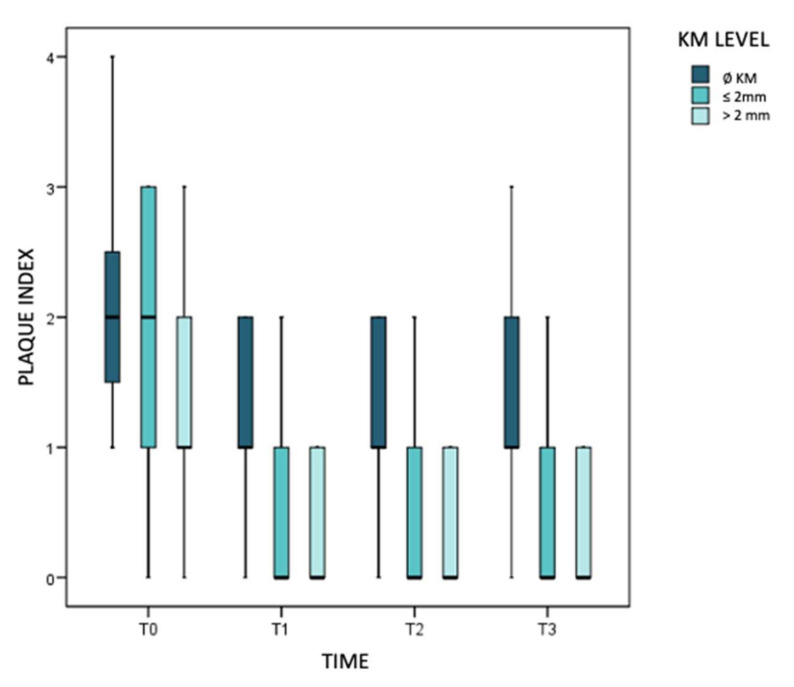
Evolution of plaque index over time.

**Table 1 jcm-11-04118-t001:** Timeline.

Timeline			
Pre-study (T0) Day 0	1 month (T1)	3 months (T2)	6 months (T3)
Baseline records and treatment	1st follow-up	2nd follow-up	3rd follow-up

**Table 2 jcm-11-04118-t002:** Clinical parameters of the implants at T0.

Clinical Parameters at T0		
	∅ KM	KM
Time since placement (years)	14	10.6
Recession (mm)	2.47 ± 1.86	0.66 ± 0.96
Bleeding %	100	100
Suppuration %	59.6	48
Probing depth (mm)	5.16 ± 1.28	4.43 ± 1.09
Bone loss %	46.78	29.97
Plaque index	2.02 ± 0.77	1.46 ± 0.89

∅ KM: absence of keratinized tissue, KM: presence of keratinized tissue.

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
