# Peer review of "The Influence of Keratinized Mucosa on the Nonsurgical Therapeutic Treatment of Peri-Implantitis"

_jcm, 2022, doi:10.3390/jcm11144118_

Round 1

Reviewer 1 Report

- The title could be changed to: The Influence of keratinized mucosa on the nonsurgical therapeutical outcomes of peri-implantitis
- The term Explantation is more appropriate than extraction of implant
- In which cases and how long after non surgical treatment is indicated surgical treatment- regeneration technique where the post operative recession is less

- In which cases surgical treatment with regeneration technique(GBR) where the post operative recession is less is a method of choice primary without first phase of nonsurgical treatment

Author Response

REVIEWER 1 RESPONSE

First of all thank you for your suggestions to improve our article, below I attach the changes we have made based on your recommendations.

  1. The title could be changed to: The Influence of keratinized mucosa on the nonsurgical therapeutical outcomes of peri-implantitis

Thank you for your suggestion. We have modified the title:

Title

The Influence of keratinized mucosa on the nonsurgical therapeutical outcomes of peri-implantitis.

  1. The term Explantation is more appropriate than extraction of implant

Thank you for pointing this out. We have modified it in the text:

" At one-month follow-up (T1), three implants had been explanted due to poor prognosis and evolution. From this moment on, the entire analysis was performed on 97 implants. "

  1. In which cases and how long after non surgical treatment is indicated surgical treatment- regeneration technique where the post operative recession is less

Thank you for your considerations, we have added it to the text:

"In the reevaluation, approximately one month from the initial decontamination treatment, it is decided if the implant continues with supportive therapy or if a surgical phase should be done because there is no control of the infection signs. If they reappear during the periodontal implant support therapy, retreatment should be considered."

  1. In which cases surgical treatment with regeneration technique(GBR) where the post operative recession is less is a method of choice primary without first phase of nonsurgical treatment.

Thank you for your considerations. The choice of a regenerative technique will depend on the characteristics of the patient and the conditions surrounding the implant (Rocuzzo 2018). Non-surgical therapy should always be performed before any surgical intervention as it gives time to clinician to evaluate the healing response of the tissue as well as the patient’s ability to perform effective oral hygiene measures (Renvert 2018).  We have added it to the text

Reviewer 2 Report

Manuscript ID: jcm-1795544

Title: Study on the influence of keratinized mucosa on the evolution of non-surgical peri-implant treatment in implants with peri-implantitis

1.What is the main question addressed by the research?

To assess the response to treatment in implants with peri-implantitis depending on the presence or absence of keratinized mucosa.

2.Is it relevant and interesting?

The article is relevant and interesting.

3.How original is the topic?

The topic is current.

4.What does it add to the subject area compared with other published material?

The authors have collected and analyzed original data.

5.Is the paper well written?

Yes, the article is well written.

6.Is the text clear and easy to read?

Moderate English editing is required.

7.Are the conclusions consistent with the evidence and arguments presented?

Yes, the conclusions consistent with the evidence and arguments presented.

8.Do they address the main question posed?

Yes, the Authors addressed the main question posed.

Other comments:

·         English language: Moderate English editing is required.

·         Introduction: This section needs few improvements. For example, Authors may include a brief sentence at the beginning of this section regarding innovations in implant dentistry based on the following reference: <[https://doi.org/10.3390/jpm12010108]>>.

·         Materials and methods: Please better define the target of statistical analysis and please indicate the software used for analysis.

·         Results: Please better define the results of statistical analysis. Figures quality is not so good. Please use other software (e.g. GraphPad)

·         Discussion: What is the main theme that emerges from the authors' analysis? Is the non surgical peri-implantitis treatment a limitation in case of esthetic rehabilitation? Please improve.

·         Conclusion: This section has been properly prepared.

After making the indicated changes, I am available for a second round of peer review.

Thanks for the opportunity to review this manuscript.

Author Response

REVIEWER 2 RESPONSE

First of all thank you for your suggestions to improve our article, below I attach the changes we have made based on your recommendations

  • English language:Moderate English editing is required.

Thank you for pointing this out. We have re-sent the article to the translator, and she has double-checked the translation.

  • Introduction: This section needs few improvements. For example, Authors may include a brief sentence at the beginning of this section regarding innovations in implant dentistry based on the following reference: <[https://doi.org/10.3390/jpm12010108]>>.

Thank you for your suggestion. We have added it to the text:

"Implant treatment has recently become a frequent treatment option in order to replace missing dental pieces. However, over the years the presence of complications, both mechanical and biological, has been repeatedly observed. Therefore, innovative materials and technologies are an intense research topic in dentistry to improve treatment outcomes, reducing at the same time morbidity, biological, and surgical times [1]. Peri-implantitis has become a relatively frequent biological complication, with a range of 1-47% [2], and an important one as well, since it can lead to the loss of the implant. "

  • Materials and methods: Please better define the target of statistical analysis and please indicate the software used for analysis.

Thank you for your considerations, we have added it to the text:

"The main objective of the statistical analysis was to describe the clinical evolution of implants with a diagnosis of periimplantitis that receive treatment and its association with KM."

"The statistical software used was SPSS 15.0."

  • Results: Please better define the results of statistical analysis. Figures quality is not so good. Please use other software (e.g. GraphPad)

Thank you for your suggestion. We have tried to better explain the results, we have changed a mistake that we have found.and we have sent the figures in another format besides the text.

  • Discussion: What is the main theme that emerges from the authors' analysis?Is the non surgical peri-implantitis treatment a limitation in case of esthetic rehabilitation? Please improve.

Thank you for pointing this out. The main topic of discussion is to point out that according to our results, KM not only has an aesthetic role around the implants, but also seems to be a protective factor, making the results of non-surgical treatment of peri-implantitis more stable in the medium term. We have added several sentences to the discussion to clarify it.

Round 2

Reviewer 2 Report

After the changes made the article may be suitable for publication.

Author Response

Thank you for your suggestions to improve our article, we have reviewed it.